# Association of Parental Socioeconomic Status and Physical Activity with Development of Arterial Stiffness in Prepubertal Children

**DOI:** 10.3390/ijerph18158227

**Published:** 2021-08-03

**Authors:** Giulia Lona, Christoph Hauser, Svea Bade, Sabrina Köchli, Denis Infanger, Katharina Endes, Oliver Faude, Henner Hanssen

**Affiliations:** Department of Sport, Exercise and Health, Medical Faculty, University of Basel, 4052 Basel, Switzerland; giulia.lona@unibas.ch (G.L.); christoph.hauser@unibas.ch (C.H.); svea.bade@unibas.ch (S.B.); sabrina.koechli@unibas.ch (S.K.); denis.infanger@unibas.ch (D.I.); katharina.endes@unibas.ch (K.E.); oliver.faude@unibas.ch (O.F.)

**Keywords:** parental lifestyle, socioeconomic status, migration background, arterial stiffness, childhood health

## Abstract

The present study examined the prospective association of parental household income, education level, migration background, and physical activity (PA) behavior with the development of pulse wave velocity (PWV) in prepubertal children. A total of 223 children (initial age 6–8 years) were included in this prospective school-based cohort study from 2014 to 2018. Parental socioeconomic status, migration background, and PA behavior were assessed by the use of questionnaires at both times points. PWV was measured by an oscillometric device at follow-up (2018). No significant association of household income, education level, and parental migration background with PWV in children after four years was found. However, a high level of maternal PA was related to a lower childhood PWV at follow-up (mean (95% CI) 4.6 (4.54–4.66) m/s) compared to children of mothers with a low PA behavior (mean (95% CI) 4.7 (4.64–4.77) m/s) (*p* = 0.049). Children of mothers with a high PA level revealed a beneficial arterial stiffness after four years. Little evidence for an association of socioeconomic status and migration background with childhood arterial stiffness was found. Increased parental PA seems to support the development of childhood vascular health and should be considered in the generation of future primary prevention strategies of childhood cardiovascular health.

## 1. Introduction

Cardiovascular disease (CVD) is the leading cause of mortality globally, accounting for 18 million deaths annually [1]. Low socioeconomic status (SES) is strongly associated with an increased risk of cardiovascular morbidity and mortality [2,3]. The burden of CVD in people with a low SES is attributable to a constellation of behavioral and psychosocial risk factors that are more prevalent in disadvantaged individuals [4]. Social disparities and migration background seem to influence cardiovascular health status at an early age [5,6,7,8]. There is evidence that childhood obesity and high blood pressure (BP) are related to a low income and education level in high-income countries [5,6]. The prospective ABCD (Amsterdam Born Children and their Development) cohort study demonstrated that children of low-educated mothers were 1.8 times more likely to have elevated BP compared to children of high-educated mothers [8]. Moreover, a low SES in childhood is inversely associated with adult body mass index (BMI) independent of adult SES, suggesting that socioeconomic inequalities and the risk of obesity tracks across the life course [9]. Children of families with low educational attainment tend to spend less time physically active [10,11], consume less healthy food [12], and be more often exposed to cigarette smoke [13]. A prospective cohort study from Sweden demonstrated that the likelihood of both being physically inactive and overweight was 30% higher in children with parental migration background compared to those of Swedish parents [7]. However, little is known about the influence of SES, migration background, and parental lifestyle on childhood vascular health. 

The assessment of arterial stiffness by pulse wave velocity (PWV) has become an established surrogate biomarker to estimate cardiovascular risk in the large arteries [14,15]. In adults, PWV is a predictor for coronary heart disease and stroke independent from risk factors such as systolic BP, cholesterol, smoking status, and the presence of diabetes mellitus [15]. A rise in 1 m/s of PWV is associated with a 23% increased risk of CVD mortality in the elderly [16]. Children with obesity or high BP have an increased PWV compared to lean children with a normal BP [17]. By contrast, higher levels of vigorous physical activity (PA) and cardiorespiratory fitness are associated with lower PWV [17,18,19]. In adults, it is known that low educational attainment is associated with higher arterial stiffness in middle-aged men and women [20]. Furthermore, a low household income is related to a PWV progression of 0.58 m/s over five years compared to participants with a high-income level, indicating a 6% increased CVD risk [21]. In young children, evidence for a prospective association between SES, migration background, and parental lifestyle with arterial stiffness remains scarce. Children of families with a high household income had a lower PWV compared to children of families with a low household income [22]. On the contrary, parental education level seems not directly to affect PWV in 6-year-old children [22,23]. However, socioeconomic factors, such as household income, occupational status, education level, and migration are often explained by or mediated through other socioeconomic indicators and influence cardiovascular health in children and adults [24]. This study aimed to analyze the prospective association of parental household income, education level, migration background, and PA behavior with PWV in prepubertal children. 

## 2. Materials and Methods

### 2.1. Study Design and Participants

The Sportcheck Follow-up Study is a longitudinal cohort study of 223 school children based in Basel City, Switzerland and set up in 2014 [25]. The study was performed in cooperation with the Department of Education of Basel City, Switzerland. All children (*n* = 1255) of the first primary school class from Basel City (26 schools) were invited to take part in the study. The children were 6–8 years old at baseline and have been followed up with a medical- and fitness screening in 2018. Written informed consent for study participation was obtained from the children and their parents at both time points. The medical and physical fitness assessments took place on separate days on-site regular school settings. The measurement of arterial stiffness was implemented in 2018. The parents of the children were requested to fill out a questionnaire about their SES, lifestyle behavior, and childhood health. Detailed information about the cross-sectional assessment is provided in the publication of baseline data [11]. The study was authorized by the ethics committee of northwest and central Switzerland (EKNZ, No. 258/12) and registered in a clinical trials registry (URL: http://www.clinicaltrials.gov: NCT03085498, accessed date 2 August 2021).

### 2.2. Measurements

#### 2.2.1. Central Arterial Stiffness

Central PWV was assessed by using a non-invasive and valid oscillometric Mobil-O-Graph device [26,27,28,29]. PWV is estimated by a transfer function that uses brachial cuff-based waveform readings and is in good agreement with the conventional method of applanation tonometry [27]. The cuff of the Mobil-O-Graph was placed on the right upper arm using appropriate cuff size for children. The measurement was performed in a sitting position after a rest of 5 min to allow for central systolic BP calibration. A minimum of two readings with good quality were obtained and averaged for the analysis. 

#### 2.2.2. Anthropometric Parameters and Physical Fitness

Anthropometric measurements, including height and weight, were assessed by a wall-mounted stadiometer (Seca, Basel, Switzerland) and bioelectrical impedance analyzer (InBody 170 Biospace devise; Inbody Co., Soul, Korea). BMI categories were applied according to the German Health Interview and Examination Survey for Children and Adolescents (KiGGS) reference values [30]. The cutoff values for overweight and obesity were determined above the 85th percentile and above the 95th percentile, respectively [30]. BP was measured with an oscillometric device (Oscillomate 9002, CAS Medical Systems, Branford, CT, USA) according to standardized procedures for children [31,32]. The same device was previously applied in a school-based intervention study (KISS) [33]. Moreover, the appropriate cuff sizes for children were used according to the guidelines [31,32]. The same device was used in the school-settings at baseline in 2014 and at follow-up in 2018 to ensure the standardization of intraindividual BP changes over time. Five readings were obtained, and the mean of the three measurements with the smallest variation was analyzed to ensure high accuracy. Furthermore, the children were classified as having normal BP (<90th percentile), elevated (≥90th percentile), or high BP (≥95th percentile) according to the KiGGS reference values [34]. In addition, a 20-m shuttle run test was performed at baseline and follow-up to assess cardiorespiratory fitness performance. The test has shown to be a valid and reliable method to estimate maximal oxygen consumption in children and adolescents [35,36]. A good interclass correlation ranging from 0.78 to 0.93 in children and adolescents was demonstrated [37]. The participants were required to run back and forth between two lines 20 m apart at a given pace marked by an audio signal. The test started with an initial running velocity of 8 km/h and progressively increased by 0.5 km/h each stage. The test ended when the participants were not able to reach the line twice in a row within the given time interval. The test result was based on the number of stages (1 stage ≙ 1 min) covered within a precision of 0.5 stages [36].

#### 2.2.3. Socioeconomic Status and Migration Background

The parental questionnaire included items of SES, migration background, and parental lifestyle, which were previously developed in a school-based intervention study in Switzerland [38]. Household income was categorized into low (under CHF 5000/month), medium (CHF 5000–9000/month), and high household income (over CHF 9000/month). Regarding parental education level, three groups of low (no vocational training), medium (vocational training or high school education), and high education (college or university degree) level of at least one parent were formed. Parental migration background was determined by the country of birth and further summarized into three groups: children of parents with a one-sided non-European migration background (one non-European parent), two-sided non-European migration background (both non-European parents), and without migration background (both parents from Europe). The parental questionnaire was translated into the seven most often spoken languages in Switzerland. 

#### 2.2.4. Parental and Family Physical Activity Behavior

In addition, we assessed PA behavior of the mother and father, as well as the PA behavior together with their child. Parents, who have reported to be never or rarely physically active were categorized as having a low PA level. Being physically active once a week was determined as a medium PA level, and all above (several times per week/daily) was determined as a high PA level. The same categorization was set for being physically active as a family. 

### 2.3. Statistical Analysis

Univariate analysis of covariate (ANCOVA) was performed to analyze the association of household income, education level, migration background, and parental PA behavior with PWVin children. The first model was adjusted for the covariates of age and sex at baseline. The additional model was extended by the covariates of household income, education level, or migration background. Furthermore, the Bonferoni post hoc method was applied to reveal the direction of the results. The results are presented by the means and its 95% confidence intervals with a two-sided level of significance of *p* = 0.005. A sample size calculation was conducted a priori based on the previously reported association between BP and retinal vessel diameters [39]. A sample of 250 children revealed a power of about 95% [39]. All analyses were performed by the statistical software Stata (Version 15, StataCorp LP, College Station, TX, USA).

## 3. Results

### 3.1. Population Characteristics

The flow diagram of the recruitment and enrollment process is shown in Figure 1. In brief, 1255 children were invited to take part in the study. In total, 540 children had written parental consent for study participation. Of these, 149 children were ill at the day of examination or relocated and 39 children did not return the parental questionnaire. Finally, 352 children had a complete baseline data set. Four years later, the same children were reexamined, and 223 (73%) children had complete parental questionnaire, including PWV measurement at the age of 10–12 years. Study participation from all schools and classes was achieved. The population characteristics are summarized in Table 1. The participants developed a significantly higher BMI (Δ 1.8 ± 1.6 kg/m^2^), BP (systolic: Δ 3.9 ± 8.2 mmHg, diastolic Δ 1.8 ± 7.6 mmHg), and cardiorespiratory fitness performance (1.9 ± 1.8 stages) after 4 years follow-up, which corresponds to an age-related normal development. The prevalence of overweight increased by 2% along with no changes in obesity from baseline to follow-up. Only one child was classified as severe underweight, and two children were classified as moderate underweight at baseline. Furthermore, a decreasing trend in elevated and high systolic (−7%) as well as diastolic BP (−5%) was observed from baseline to follow-up. The 10–12-year-old children had a mean PWV of 4.7 m/s (±0.3) on average. Regarding the development of household income over four years, 12.3% of the families had a higher income at follow-up and respectively, 4.9% of the families had a lower household income at follow-up.

### 3.2. Association of Socioeconomic Status and Migration Background with Childhood Arterial Stiffness 

The association between baseline household income, education level, and migration background of the parents with arterial stiffness in children is shown in Table 2. In our cohort, no significant differences in childhood PWV between the groups of low (mean 4.67 m/s, *n* = 40), medium (mean 4.67 m/s, *n* = 70), or high (mean 4.67 m/s, *n* = 95) household income (*p* = 0.573) were detected. A slightly lower mean PWV was observed in children with parents of a high education level (mean 4.6 m/s, *n* = 141) compared to a low (mean 4.68 m/s, *n* = 74) or medium (mean 4.72 m/s, *n* = 7) education level. However, this association was not statistically significant (*p* = 0.554). Furthermore, we found little evidence for an association between migration background and childhood PWV between the groups of parents with (one-sided: mean 4.71 m/s, *n* = 25 and two-sided: mean 4.72 m/s, *n* = 9) or without (mean 4.64 m/s, *n* = 184) European nationality (*p* = 0.445).

### 3.3. Association of Parental and Familial Physical Activity Behavior with Childhood Arterial Stiffness

Table 3 summarizes the findings of the association between parental and familial PA with arterial stiffness in children. Children of mothers with a high PA level showed a significantly lower PWV (mean 4.6 m/s, *n* = 82) compared to children of mothers with a low PA behavior (mean 4.7 m/s, *n* = 67) (*p* = 0.049). No significant association between child PWV and the three categories of low (mean 4.69 m/s, *n* = 67), medium (mean 4.64 m/s, *n* = 59), and high (mean 4.65 m/s, *n* = 101) PA behavior of the father was found (*p* = 0.483). Regarding the relationship of being physically active as a family, a trend toward a lower childhood PWV in the most active families (mean 4.66 m/s, *n* = 91) was detected compared to the families with a medium (mean 4.62 m/s, *n* = 81) or low (mean 4.73 m/s, *n* = 43) PA behavior, however, without statistical significance (*p* = 0.108). 

## 4. Discussion

Our study results demonstrate that maternal PA behavior influences arterial stiffness in children over four years. Children of mothers with a high PA level had significantly lower PWV at follow-up compared to children of mothers with a low PA level. Furthermore, we found little evidence for a relationship between parental SES and migration background with the development of childhood arterial stiffness. 

Our data indicate no significant prospective association of parental education level, household income, or migration background with PWV in children. In another prepubertal child cohort, we have previously shown that household income and parental educational level are cross-sectionally associated with PWV in children [22]. However, these findings were not confirmed by the Generation R study, revealing little evidence for an inverse relationship between maternal education and arterial stiffness in six-year-old children [23]. Regarding the prospective association, the Young Finn study concluded that a high parental income in childhood was associated with a favorable PWV in young adulthood compared to children of parents with a low household income [40]. However, this association remained non-significant after adjusting for adult cardiometabolic risk factors [40]. The longitudinal Whitehall II study demonstrated that SES was inversely associated with arterial stiffness progression in middle-aged adults over five years independent of smoking status, alcohol consumption, and BP [21]. In our cohort, SES did not seem to affect arterial stiffness in children over four years. Our sample consisted of a large group of well-educated (63%) and wealthy (46%) parents with a low migration background. In this respect, one needs to be cautious to generalize our results to other countries and cities with lower education and wealth status. Furthermore, a short lifetime exposure to social disparities and cardiovascular risk has to be considered in these young children. As described above, there is good evidence that the relationship between vascular impairments and SES may develop over time. The study by Lam et al. demonstrated that the magnitude of the association of low-grade inflammatory markers with SES increases progressively with age [41]. The authors assumed a cumulative effect of social inequalities on inflammation level across the life course [41]. Impairments of arterial stiffness in children of socially disadvantaged families might manifest at later time points. Furthermore, large artery stiffness is characterized in large parts by structural remodeling and less so by functional components such as endothelial function. To detect changes in PWV, more intense stimuli and longer-term exposure appear to be necessary. In addition to the most likely explanation for the lack of association being short-term exposure, family background might be of growing importance in further education and secondary school when children are categorized by their school performance. Especially schools at lower educational levels may need to implement health intervention programs focusing more on familial PA and lifestyle behavior. 

Parental SES can hardly be adjusted, but parental lifestyle behavior is adaptable. Therefore, we assessed parental PA behavior and time spent physically active as a family in relation to the development of childhood arterial stiffness. Our study results indicate that children of mothers with a high PA level have a significantly lower arterial stiffness compared to children of mothers with medium or low PA behavior. It has become evident that children of active mothers are twice as likely to be active as children of inactive mothers, [42] which is in line with our findings. Interestingly, this association was not observed in fathers. At a cross-sectional level, we recently demonstrated that children of physically active parents had a lower PWV compared to children of parents with low PA behavior at the age of 6–8 years [22]. Moreover, the Uppsala Family study from Sweden concluded that parental lifestyle habits had a greater impact on child cardiovascular risk factors than parental SES [43]. More research on the association between parental lifestyle and the development of vascular health in childhood is warranted to clarify these interrelations. In terms of the relationship between familial PA behavior and PWV in children, no significant prospective relationship was found. Children may spend more time with their peers at the age between 10 and 12 years, which might attenuate the impact of familial PA behavior on large vascular health in young children. 

Our findings have several practical implications. In young children with short lifetime exposure to socioeconomic and environmental risk factors, the vascular health of large arteries was not altered. It appears that longer-term exposure is necessary to establish the association of these risk factors with vascular alterations in adolescence and young adulthood [40,41]. However, especially maternal PA behavior may serve as a role model very early in life in order to guide children into a healthy lifestyle to maintain vascular health during childhood development. As a consequence of our findings, the knowledge of the importance of parental PA behavior needs to be disseminated into school educational programs and caretakers of family heath. Our results support the belief that school intervention programs need to target the families rather than merely the children at schools [44]. Parents need to be made aware of their status as a role model with respect to PA behavior and associated cardiovascular health.

Our findings need to be interpreted in light of some limitations. The lost-to follow-up rate was 27% due to illness at the day of examination, relocation, or personal reasons. However, differences in the baseline characteristics between the follow-up and non-follow-up participants were small and thus, the present cohort can be considered as representative of the original population [39]. Another limitation is that PWV was only assessed in 2018, and therefore, adverse causality cannot be ruled out. We did not control for pubertal status, which may bias our results. However, a recent systematic review and meta-regression has demonstrated that PWV progresses linearly from childhood to adolescence without marked changes at the onset of puberty in girls and boys [45]. To determine the impact of other cardiovascular risk factors such as hypercholesterolemia on childhood PWV, further research is warranted in children with manifest disease, as the prevalence of risk factors is low in a population-based cohort of young children. The parental questionnaire was translated into the seven most often spoken languages in Switzerland. Other spoken languages could not be taken into account, which might have resulted in unanswered or incorrect completed parental questionnaires. Furthermore, we cannot rule out that completion of the questionnaire may have been affected by educational level. The answers of the questionnaire items were assessed by categories. Absolute values might have resulted in a better differentiation in PWV. This study is also limited in its variation in SES among parents, since most families had a fairly high education, standard of living, and a low migration background. Last but not least, parental smoking status was not reported at baseline and might further bias our results, considering that smoking status is an important determinant in the development of child vascular health [46].

## 5. Conclusions 

A higher maternal PA level was related to a favorable development of arterial stiffness in children over four years. Little evidence for a prospective association of SES and parental migration background with arterial stiffness in children was found. Parental lifestyle, in particular maternal PA, seems to be a key part of the exposure that affects the development of childhood arterial stiffness. However, more longitudinal studies with follow-up into adulthood and potential manifest CVD are warranted to determine the clinical relevance of social disparities and parental lifestyle behavior on arterial stiffness progression in childhood. Nevertheless, our results indicate the importance of an active parental lifestyle to achieve good vascular health in childhood as a preventive measure to reduce CVD development later in life.

## Figures and Tables

**Figure 1 ijerph-18-08227-f001:**
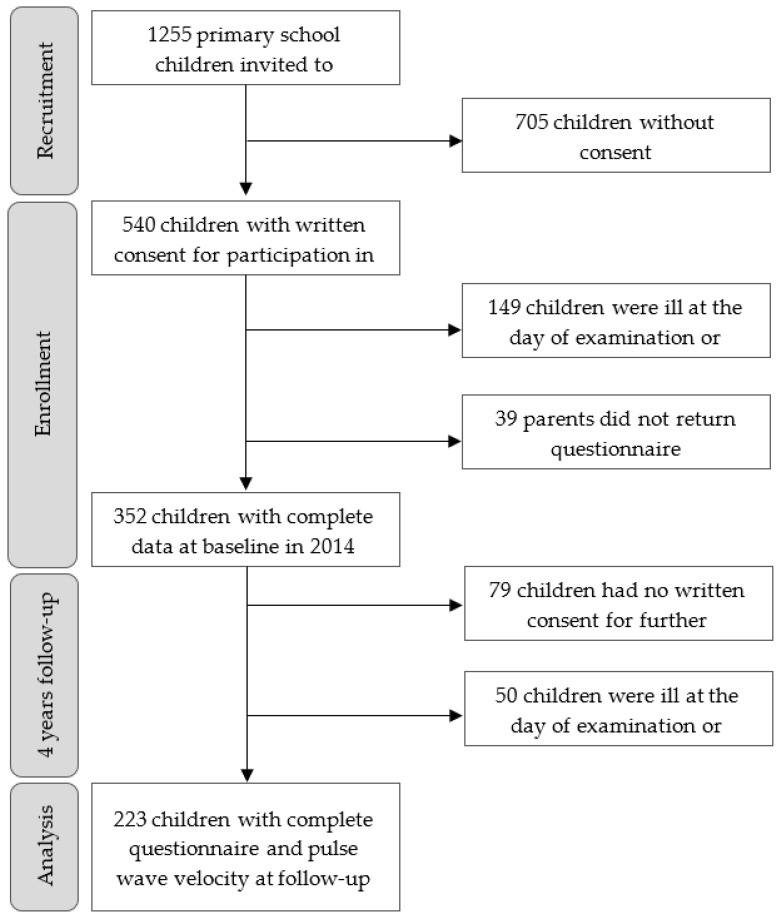
Flowchart.

**Table 1 ijerph-18-08227-t001:** Population characteristics.

		Baseline		4 Years Follow-Up		Difference		
	*n*	Mean	SD	Mean	SD	Mean	SD	*p*-Value
Sex (male, %)	223	47						
Age (y)	223	7.4	0.3	11.4	0.3			
Height (cm)	210	125.9	5.1	149.0	6.6	23.0	3.6	<0.001
Weight (m)	210	25.4	3.9	40.0	7.7	14.3	4.9	<0.001
BMI (kg/m^2^) Normal (%) Overweight (%) Obese (%)	210	16.01 92 6 2	1.8	17.8 90 8 2	2.8	1.8	1.6	<0.001
Z-BMI *	210	-0.15	0.9	-0.36	0.5	0.2	0.6	<0.001
Systolic BP (mmHg) Normal (%) Elevated (%) High (%)	210	104.1 83 10 7	7.7	108.0 90 5 5	7.8	3.9	8.2	<0.001
Z-systolic BP *	210	0.56	0.93	1.5	0.9	1.0	1.02	<0.001
Diastolic BP (mmHg) Normal (%) Elevated (%) High (%)	210	65.2 76 10 14	7.2	67.0 81 9 10	6.5	1.8	7.6	<0.001
Z-diastolic BP	210	0.56	1.1	1.3	0.9	0.74	1.1	<0.001
PWV (m/s)	223			4.7	0.3			
CRF (stages)	202	4.7	1.7	6.5	2.1	1.9	1.8	<0.001

* Z-BMI, Z-systolic BP, and Z-diastolic BP are based on the KiGGS (German Health Interview and Examination Survey for Children and Adolescents) reference values. BMI indicates body mass index; BP, blood pressure; PWV, pulse wave velocity; CRF, cardiorespiratory fitness (1 stage ≙ 1 min); SD, standard deviation.

**Table 2 ijerph-18-08227-t002:** Childhood pulse wave velocity in relation to categories of parental household income, education level, and migration background.

Parameter	Model	*n*	Pulse Wave Velocity at Follow-Up (Increase per 1 m/s)	
			Mean (95% CI)	*p*-Value
**Household income at baseline**				
Low (<CHF 5000/month)	1	40	4.56 (4.43 to 4.68)	0.273
Medium (CHF 5000–9000/month)		70	4.66 (4.57 to 4.75)	
High (>CHF 9000/month)		95	4.69 (4.58 to 4.79)	
Low (<CHF 5000/month)	2	40	4.6 (4.51 to 4.69)	0.382
Medium (CHF 5000–9000/month)		70	4.67 (4.6 to 4.73)	
High (>CHF 9000/month)		95	4.68 (4.62 to 4.74)	
**Education level at baseline**				
Low (no vocational training)	3	7	4.72 (4.52 to 4.93)	0.544
Medium (vocational training/high school education)		74	4.68 (4.62 to 4.75)	
High (college/university degree)		141	4.64 (4.6 to 4.69)	
Low (no vocational training)	4	7	4.77 (4.54 to 5)	0.466
Medium (vocational training/high school education)		74	4.68 (4.61 to 4.75)	
High (college/university degree)		141	4.64 (4.59 to 4.69)	
**Migration background at baseline**				
European	2	184	4.64 (4.6 to 4.68)	0.445
One-sided non-European		25	4.71 (4.6 to 4.81)	
Two-sided non-European		9	4.72 (4.54 to 4.9)	
European	5	184	4.64 (4.6 to 4.68)	0.198
One-sided non-European		25	4.72 (4.6 to 4.83)	
Two-sided non-European		9	4.79 (4.59 to 4.99)	

Model 1: adjusted for age, sex, and household income at follow-up. Model 2: adjusted for age and sex. Model 3 adjusted for model 2 plus education level. Model 4: adjusted for model 2 plus household income. Model 5: adjusted for model 1 plus education level and household income.

**Table 3 ijerph-18-08227-t003:** Childhood pulse wave velocity in relation of categories of parental and familial physical activity behavior.

Parameter			Pulse Wave Velocity at Follow-Up (Increase per 1 m/s)	
	Model	*n*	Mean (95% CI)	*p*-Value
**Maternal physical activity level at baseline**				
Low (<1/week)	1	67	4.7 (4.64 to 4.77)	0.049
Medium (1/week)		73	4.69 (4.62 to 4.75)	
High (>1/week)		82	4.6 (4.54 to 4.66)	
**Paternal physical activity level at baseline**				
Low (<1/week)	1	62	4.69 (4.62 to 4.76)	0.483
Medium (1/week)		59	4.64 (4.56 to 4.71)	
High (>1/week)		101	4.65 (4.6 to 4.71)	
**Familial physical activity level at baseline**				
Low (<1/week)	1	43	4.73 (4.65 to 4.82)	0.108
Medium (1/week)		87	4.62 (4.57 to 4.68)	
High (>1/week)		91	4.66 (4.6 to 4.72)	

Model 1: adjusted for age and sex.

## Data Availability

The data presented in this study are available on request from the corresponding author.

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
