# Peer review of "Association of Parental Socioeconomic Status and Physical Activity with Development of Arterial Stiffness in Prepubertal Children"

_ijerph, 2021, doi:10.3390/ijerph18158227_

Round 1

Reviewer 1 Report

Review of the manuscript of Lona et al. “Association of parental socioeconomic status and physical activity with development of arterial stiffness in prepubertal children.”

The manuscript supplements novel knowledge in an important field of research, the reduction of cardiovascular risk factors in childhood in order to prevent cardiovascular events in adulthood. Therefore, the manuscript seems to be appropriate for publication in IJERPH.

However, I have some minor comments:

  1. If the authors include 8 years old girls at baseline and 4 years later 12 years old girls at the follow-up examination, many of these girls will be pubertal at the follow-up examination. In contrast, boys mostly enter into puberty later in life. Do puberty will influence PWV? Do you see gender and age differences?
  2. Investigating young children, examined differences in PWV are always rather small. Do you believe that the consideration of more cardiovascular risk factors like hypercholesterolemia, lipoprotein (a) elevations or passive smoking would lead to a more powerful differentiation?
  3. The Flow chart of figure 1 needs correction.

Reviewer 2 Report

Based on 223 children as samples, this paper analyzes the relationship between family income, education level, migration background, parents' sports behavior and children's pulse wave velocity by using the method of univariate analysis of covariance (ANCOVA). The results show that: there is a significant negative correlation between mother's sports behavior and children's pulse wave velocity, and there is no significant prospective correlation between parents' education level, family income or immigration background and children's pulse wave velocity. After review, there are the following problems in the paper

1.The article lacks practical significance. This paper fails to elaborate the practical significance of the research conclusion. For example, suggestions on heart disease diagnosis, sports training program or health management methods of physical examination enterprises.

2.The theoretical basis of the relationship between variables is not solid, and there is a lack of a unified theoretical framework to link the variables.

3. The discussion and analysis of the research results are not thorough enough. The results showed that education level, family income or immigration background had no significant prospective correlation with children's pulse wave velocity. But the reason is not analyzed in theory, which also leads to a great discount of research value.

In view of the above reasons, this paper has not yet met the publishing requirements of the journal. It is suggested to explore the research significance of the article in combination with practice, elaborate the theoretical basis in detail, and make innovation explanation on this basis.

Reviewer 3 Report

  1. Reliability and validity of anthropometric measurements missing.
  2. The reliability of 20m shuttle run also not well articulated. Furthermore, one wonders if the shuttle run was don at baseline and also in the follow up measurements.
  3. It is not clear in the study if the no European migration parents questionnaire were translated to their native language.
  4. In the parental and family PA behaviour one wonders as to what the educational level affect the completion of the PA questionnaire which is based on the tie sequences.
  5. It will be interesting for the other readers how the 1255 children were recruited including the geographical setting of the sample.
  6. The sample size in figure 1 and table 1 is confusing given the 210 sample and the 223 sample. Furthermore, it will be interesting to some readers how the missing data were handled in the study.
  7. Given the migration characteristics nature of the general population across the globe and the low household income at base line of the study , one wonders the level of undernutrition if it does exist in the sample at baseline and at follow-up measurements.
  8. In the discussion section, the statement “ our sample was homogenous consisting of a large group of well educated and wealthy parents”  is not consistent with the information on table 2. However, the heterogeneity/homogeneity of the sample from baseline up to four years could be of interest to some readers.
  9. In the period of four years from baseline  one wonders if the distribution of household income between sample did not change and how the author accounted for these changes in the analysis could interest some readers.

Reviewer 4 Report

Thank you for the opportunity to review your manuscript.  It is an interesting study. I hope parents read your article and realize how important their activity behavior can influence their children's health.

I have a few suggestions to improve the manuscript.  On the bottom of page 3, under Population characteristics, in the sentence, "The population characteristics is summarized...," there is a grammatical error. The subject of the sentence is "characteristics" which requires the verb to be in its plural form (i.e., are). On page 7, in the first sentence in the second paragraph, "Our data indicates," the word DATA requires the verb to be in the plural form because DATA is the plural form of DATUM.  Thus, your sentence should read, "Our data indicate..."
